# Investigation of the Physical Mechanical Properties and Durability of Sustainable Ultra-High Performance Concrete with Recycled Waste Glass

**Mohamed Amin** [1,2], **Ibrahim Saad Agwa** [1], **Nuha Mashaan** [3,*], **Shaker Mahmood** [4,5,*] and **Mahmoud H. Abd-Elrahman** [6]

1   Civil and Architectural Constructions Department, Faculty of Technology and Education, Suez University, Suez 43721, Egypt
2   Civil Engineering Department, Mansoura High Institute for Engineering and Technology, Mansoura 35516, Egypt
3   Department of Civil Engineering, School of Civil and Mechanical Engineering, Curtin University, Bentley, WA 6102, Australia
4   Department of Civil Engineering, College of Engineering, University of Duhok, Duhok 42001, Iraq
5   Department of Civil Engineering, College of Engineering, Nawroz University, Duhok 42001, Iraq
6   Civil Engineering Department, El-Arish High Institute for Engineering and Technology, EL-Arish 45511, Egypt
*   Correspondence: nuhas.mashaan1@curtin.edu.au (N.M.); shaker121090@gmail.com (S.M.)

**Abstract:** Construction material sustainability and waste reuse have emerged as significant environmental issues. Concrete is widely used in the building and engineering fields. Ultra-high performance concrete (UHPC), which has remarkably high mechanical properties, has become one of the most common concrete varieties in recent years. As a result, substantial amounts of Portland cement (PC) are frequently used, raising the initial cost of UHPC and restricting its broad use in structural applications. A significant amount of $CO_2$ is produced and a large amount of natural resources are consumed in its production. To make UHPC production more eco-friendly and economically viable, it is advised that the PC in concrete preparations be replaced with different additives and that the recycled aggregates from various sources be substituted for natural aggregates. This research aims to develop an environmentally friendly and cost-effective UHPC by using glass waste (GW) of various sizes as an alternative to PC with replacement ratios of 0%, 10%, 20%, 30%, 40%, and 50% utilizing glass powder (GP). Fine aggregate "sand (S)" is also replaced by glass particles (G) with replacement ratios of 0%, 50%, and 100%. To accomplish this, 18 mixes, separated into three groups, are made and examined experimentally. Slump flow, mechanical properties, water permeability, and microstructural characteristics are all studied. According to the results, increasing the S replacement ratio with G improved workability. Furthermore, the ideal replacement ratios for replacing PC with GP and S with G to achieve high mechanical properties were 20% and 0%, respectively. Increasing the replacement rate of GP in place of PC at a fixed ratio of G to S resulted in a significant decrease in water permeability values. Finally, a microstructural analysis confirms the experimental findings. In addition, PC100-S100 was the best mix compared to PC100-S50 G50 and PC100-G100.

**Keywords:** ultra-high performance concrete; glass powder; sustainability; mechanical characteristics; microstructure



## 1. Introduction

The sustainability of construction materials and trash reuse have become essential environmental issues [1,2]. Concrete is extensively employed in construction and engineering and is a significant material in many building types [3,4]. It comprises many materials that join together to provide necessary strength [3] and performs well in compressive strength [5–7]. One of the varieties of concrete that has become more popular over the

last several decades is known as "ultra-high performance concrete", distinguished by extremely high mechanical characteristics and remarkable durability [8–11] and may be considered a new type of construction material [12–14]. It may be utilized in a variety of important projects, including offshore platforms, urban furniture, decking, and concrete restoration [15–19]. The constituents of UHPC are high density Portland cement (PC), steel fibers, aggregates, a high-range water reducer, and a minimal quantity of water [20,21]. UHPC is made by boosting the PC content's density to the maximum achievable degree. Thus, large PC volumes are often employed to create a dense PC content; UHPC mixtures have a PC dose of 900~1100 kg/m$^3$ [22], roughly two to three times more than conventional concrete. Owing to the high PC content in UHPC, the material's original cost has risen, limiting its widespread usage in structural applications.

Furthermore, high PC volumes generate a considerable quantity of carbon dioxide. Enormous amounts of resources and energy have been used, which led to an increase in the possible environmental repercussions of UHPC on a global scale [23]. Thus, to enable the building industry to be more ecologically and economically sustainable, it is recommended to replace the PC in concrete preparations with various additives and to substitute diverse sources of natural aggregates with recycled aggregates.

The utilization of construction and demolition waste (CDW) as a PC replacement or natural aggregate replacement is desirable to achieve this goal [24–26]. Glass trash, which is utilized in common things, such as bottles, candles, screens, and windshields, is one of the CDWs whose use has recently expanded. It contributes significantly to the waste produced by cities, and the amount of glass trash has grown exponentially over the last several years [27,28]. According to government data, the daily average glass waste (GW) weight in recent years was approximately 300 tons [29–31]. These glass components must be substituted in concrete to prevent these issues. One significant approach to recycling glass trash is utilizing it in PC products. A considerable quantity of glass trash may be recycled in the building sector [32,33] by substituting PC or aggregates with recycled glass. It may diminish the environmental impact by recycling an otherwise useless item and lowering PC production carbon emissions [34]. Glass has a high enough percentage of silica to be used instead of fine aggregate [20,29,35]. This lessens the GW that pollutes landfills and might lessen the amount of river sand required for construction, which would bring down the cost of installation [36,37]. Shattered glass of less than 75 microns in size may also be used as a PC substitute in applications where pozzolanic properties are desired. The silica in glass may be transformed into calcium silicate hydrate (C–S–H) as a result of the calcium hydrate reaction (Ca(OH)$_2$) [38,39]. It has additionally been observed that glass will begin to display pozzolanic behaviors if the diameter is less than 150 microns; moreover, the pozzolanic behaviors will improve if the particle diameter is reduced to less than 35 microns, leading to better outcomes than 150 micron-sized particles [40]. In brief, according to the findings of several researchers, trash may be effectively used as a binding ingredient in concrete [41–43] and GW may be reused in concrete without compromising its quality in any way [44], but while also improving the mechanical performance of concrete [45] and enhancing its durability [46–48].

Several kinds of research have appeared in publications to explore the implications of utilizing GW as a PC alternative. According to Letelier et al. [49], an optimum particle size of 38 μm is compatible with a 30% PC (by weight) partial substitute without compromising the material properties. This is corroborated by the observations of Shao et al. [50], who noticed that a grain size of 38 μm is compatible with a 30% PC (by volume) replacement. Research by Khmiri et al. [51] corroborated this idea, indicating that although employing GW with a grain size of 40 μm diminishes the compressive strength, adding 20% from PC weight substitution of GW milled to 20 μm may enhance the mechanical characteristics. Mostofinejad et al. [37] demonstrated that the 28- and 300-day compressive strengths dropped by 40% and 42%, respectively, when PC was substituted with milled GW at a 30% level. Letelier discovered that adding 20% replacement at 38 μm exhibited the best improvement in compressive strength through 28~90 days; specimens with the same size

diameter with 10% and 30% substitutions showed a higher strength than the reference sample. It has also been discovered that the best dose involves the replacement by weight of the PC with glass powder (GP) up to 20% by Z. Ismail [52] and Tejaswi et al. [53]. Shayan [54] found that the use of scrap GP with a particle size smaller than 15 μm in place of PC resulted in a decrease in compressive strength after 28 days. This trend was seen for all increased percentages of PC substitution. Ali et al. [55] indicated that samples substituting GW at 10 μm, 20 μm, and 40 μm showed a characteristic strength of 55.5 MPa, 66.6 MPa, and 51.5 MPa, respectively, indicating that the pozzolanic activity was enhanced by using tighter GW grains. This is in agreement with Shi et al. [56], who illustrated that after 28 days, compressive strengths were greater in specimens produced with GW that had finer particle sizes than in those made with more roughly distributed particles. Z. Chen's work [57] clarified similar findings, where raising the milling period of GW massively improved the specimens' compressive strength after 7 and 28 days.

Another potential choice for GW utilization in PC products is the substitution of natural aggregate [58–61]. Due to its comparable density (2.60) and low absorption coefficient, recycled glass is employed as both coarse and fine particles in concrete [52,62,63]. Regarding mechanical characteristics, Penacho et al. [64] discovered that when 20%, 50%, or 100% glass (by volume) was used instead of S, the compressive and flexural strengths were higher. However, when 100% glass was used instead of S, the compressive strength was lower after 28 days. Gerges et al. [65] clarified that boosting the amount of fine GW used in place of S has an enormously negative impact on the mechanical characteristics [66–71]. Turgut and Yahlizade [72] illustrated that 20% supplementation of S with glass S enhanced the indirect tensile strength, compressive strength, flexural strength, and abrasion resistance of paving blocks by 47%, 69%, 90%, and 15%, respectively, versus the control. On the other hand, new results from the literature show that Mardani Aghabaglou et al. [64] and Ali and Al-Tersawy [68] have discovered a rising slump as the replacement concentration rises. The authors explain this by saying that the tiny glass grains are more easily compacted than coarser S ones. Furthermore, because glass has a reduced unit weight, the scientists discovered that specimens with greater levels of GW had reduced dry and fresh densities [73,74].

*Research Significance*

This research aims to produce an environmentally friendly and economical UHPC by utilizing GW of different sizes as a substitution of PC with substitution rates of 0%, 10%, 20%, 30%, 40%, and 50% using GP and as a substitution for the fine aggregate, with substitution ratios of 0%, 50%, and 100% using G. To achieve this goal, 18 mixtures are made and tested experimentally. Slump flow, mechanical properties, water permeability, and microstructural characteristics are investigated.

## 2. Experimental Work

### 2.1. Materials Used

Portland cement (PC) (CEM I 52.5 N) was employed in this experimental program. UHPC mixes have a large quantity of PC (1000 kg/m$^3$), as described in previous works [75,76]. The properties of PC were measured according to BSEN197/1 2011 [77].

Silica fume (SF) was obtained from SICA Company in Egypt (a by-product of ferrosilicon alloy production), and it abides by the ASTM C1240 standards [78]. Additionally, SF with sizes from 0.1 to 1 μm are essential to produce UHPC [79,80].

The GW utilized in this work was collected from local workshops in Mansoura city, Egypt. The bottles' metal or plastic taps and neck rings were then removed. The glass bottles were then carefully washed with tap water to remove any paper or plastic labels on the surface and any contamination. After washing and drying, the glass bottles were crushed into tiny particles using a jaw crusher, which met the ASTM C 33 [81] grading criteria for S, for 10 min. G that passed through 4.75 mm sieves was used. On the other hand, to obtain the GP to replace the PC, the milling time was increased for another 20 min,

and particles that passed through 75 μm sieves were utilized. Figure 1 illustrates the different stages of glass waste milling. Scanning electronic microscopy (SEM) analysis was conducted on GP, as shown in Figure 2. The physicochemical parameters of used binder materials are listed in Table 1.

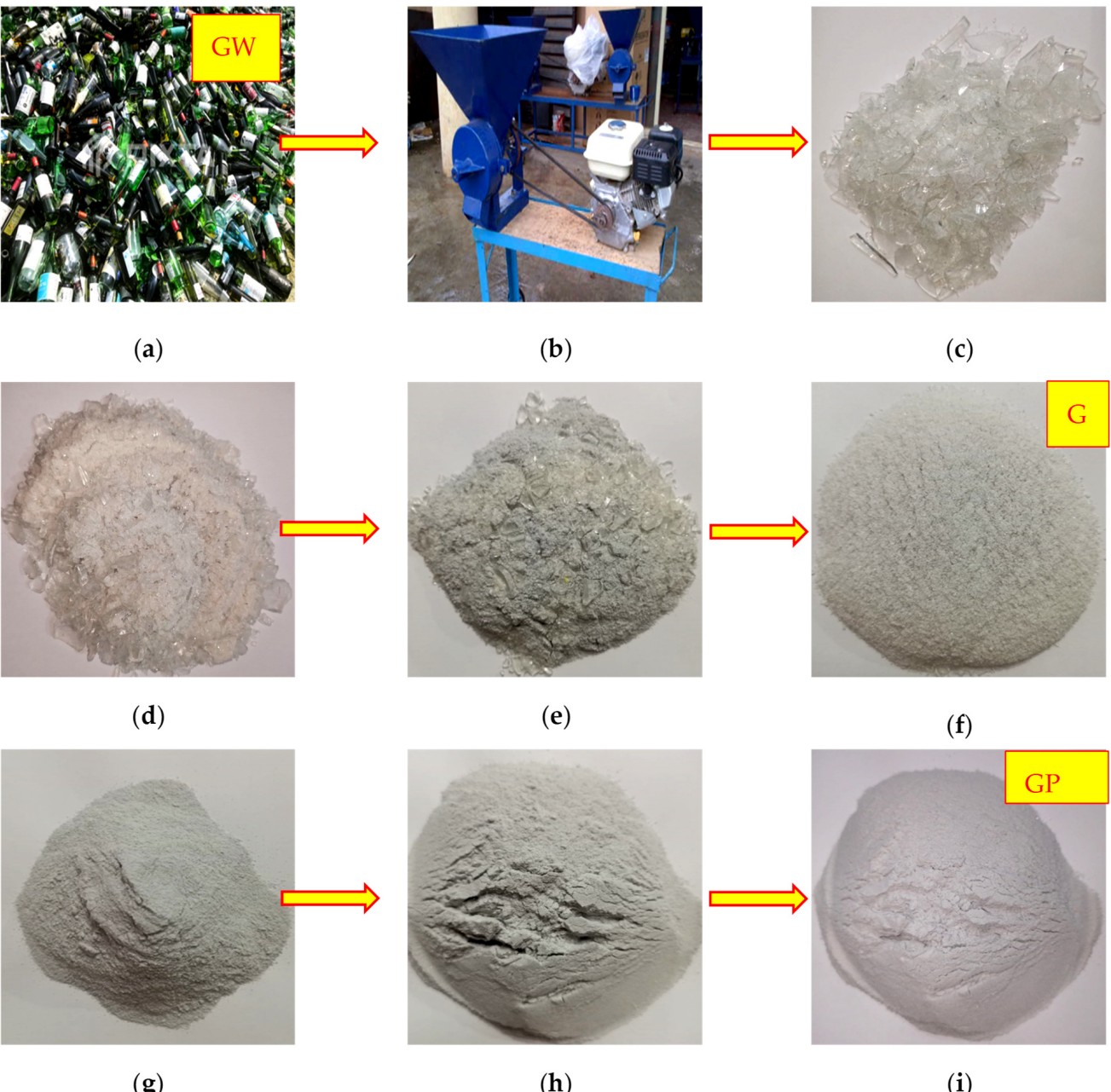

**Figure 1.** Different stages for glass waste milling. (**a**) Glass waste (GW); (**b**) ground mill; (**c**) crushed glass (size >> 4.75 mm); (**d**) crushed glass (size >> 4.75 mm); (**e**) crushed glass (size > 4.75 mm); (**f**) glass particles (G) used as fine aggregate (size < 4.75 mm); (**g**) milled Glass (size << 4.75 mm); (**h**) milled glass (size <<< 4.75 mm); and (**i**) glass powder (GP) used as a substitute for PC (size < 75 μm).

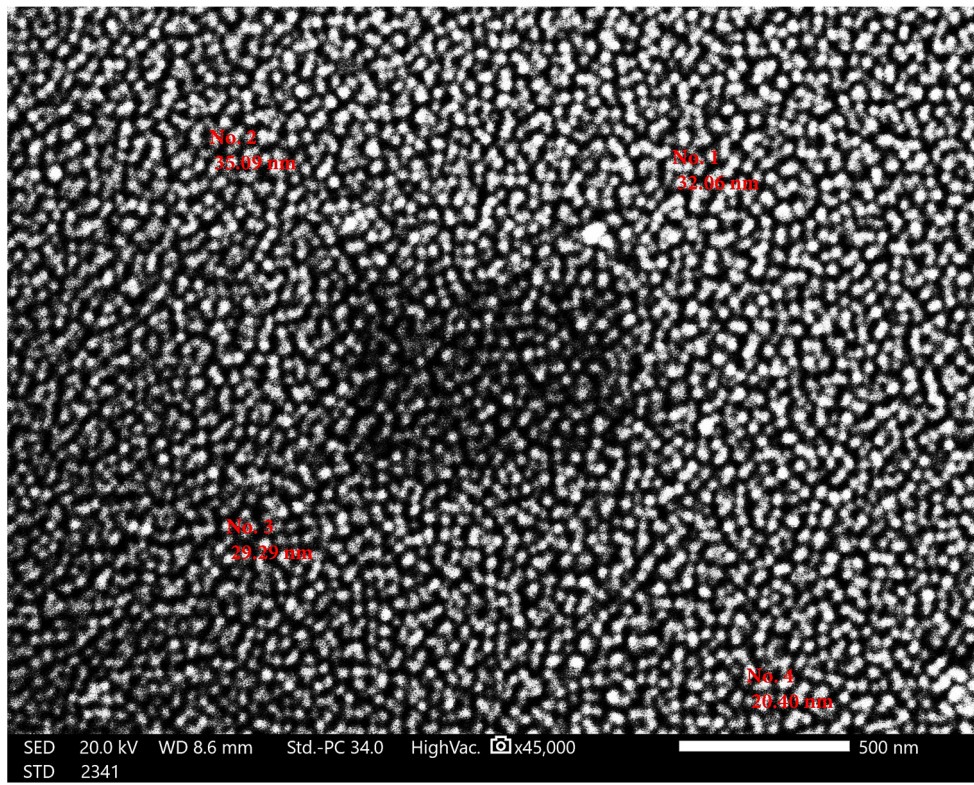

**Figure 2.** SEM micrographs of the glass powder.

**Table 1.** The physicochemical parameters of binder materials.

| Properties | Portland Cement | Silica Fume | Glass Powder |
|---|---|---|---|
| Physical | | | |
| Specific gravity | 3.15 | 2.15 | 2.60 |
| Initial setting time (min) | 71 | - | - |
| Final setting time (min) | 309 | - | - |
| Specific area ($cm^2/gm$) | 3495 | 21,160 | 5630 |
| Color | Gray | Light Gray | White |
| Chemical compositions (%) | | | |
| $SiO_2$ | 21.95 | 98.76 | 75.10 |
| $Al_2O_3$ | 3.97 | 0.27 | 1.69 |
| $Fe_2O_3$ | 4.28 | 0.28 | 0.36 |
| CaO | 60.86 | 0.15 | 10.97 |
| MgO | 4.55 | 0.14 | 0.90 |
| $SO_3$ | 2.18 | 0.13 | 0.13 |
| $K_2O$ | 0.75 | 0.14 | 0.20 |
| $Na_2O$ | 0.83 | 0.13 | 10.65 |
| LOI | 0.63 | - | - |

Quartz powder (QP) was used as a filler in this study, with particle sizes ranging from 5 to 20 μm. All UHPC mixes were designed with QP (smaller than 10 μm) [82–84]. Table 2 lists the characteristics of QP.

**Table 2.** The physicochemical parameters of quartz powder.

| Properties | Quartz Powder |
|---|---|
| Physical | |
| Specific gravity | 2.57 |
| Specific area (cm$^2$/gm) | 5520 |
| Color | White |
| Chemical compositions (%) | |
| SiO$_2$ | 97.18 |
| Al$_2$O$_3$ | 0.38 |
| Fe$_2$O$_3$ | 0.49 |
| CaO | 0.86 |
| SO$_3$ | 1.09 |

In this research, crushed granular GW was used as a fine aggregate. This helps save natural resources and reduces pollution [68,85,86]. When used as a fine aggregate, the crushed GW protects concrete from the damaging effects of chloride [87]. The GW was shattered using a hammer before being placed in a Los Angeles abrasion mill and rolled to create glass particles (G) ranging in size from 0.150 mm to 4.75 mm, which was used as a fine aggregate as a replacement for S. Testing on fine aggregates was performed according to ASTM C33/C33M-18 [88]. Table 3 summarizes the features of the fine aggregates. Figure 3 portrays the gradation curve of the fine particles utilized, while Figure 1 displays different stages for glass waste milling.

**Table 3.** Physical and mechanical characteristics of aggregates.

| Properties | Sand | Glass Particles |
|---|---|---|
| Specific gravity | 2.67 | 2.63 |
| Unit weight (kg/m$^3$) | 1695 | 1865 |
| Water absorption (%) | 0.85 | 0.65 |
| Clay and fine materials (%) | 0.47 | 0.30 |

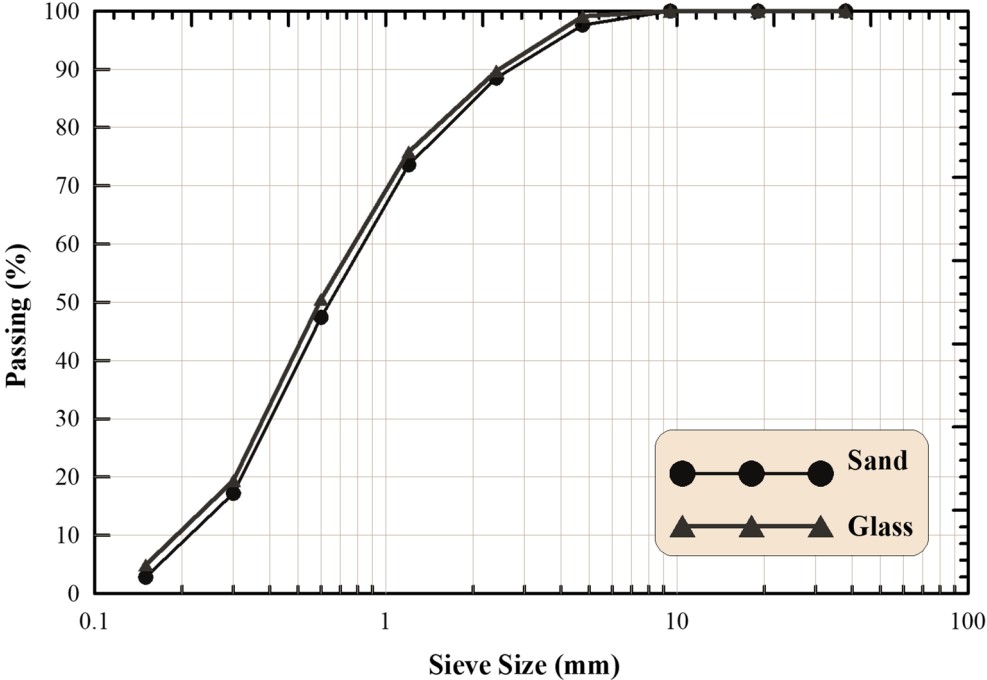

**Figure 3.** The grading curve of the fine aggregates used.

Steel fiber was employed to study UHPC reinforcement ratios. According to studies, UHPC with a steel fiber volume concentration of 1.5% is adequate [89]. The steel fiber properties provided by the manufacturer are shown in Table 4.

**Table 4.** Properties of steel fibers.

| Length (mm) | Diameter (µm) | Aspect Ratio | Modulus of Elasticity (GPa) | Density (g/cm$^3$) |
|---|---|---|---|---|
| 12 | 150 | 80.3 | 190 | 7.85 |

Finally, all UHPC mixes had a 2.2% superplasticizer (SP) concentration in the binder ingredients. ASTM C-494 Type G fulfills the SP standards with a specific gravity of 1.08 and a clear liquid color [90] and BS EN 934 part 2:2001 [91].

### 2.2. Research Methodology

The experimental work was carried out on three main groups with a total number of 18 UHPC mixes and based on the absolute volume method to satisfy the goals of this article. As shown in Table 5, each group comprises six mixtures with a single partial substitution of the PC weight at levels of 0, 10, 20, 30, 40, and 50%. The first group included only 100% S as a fine aggregate with a PC content of 500 to 1000 kg/m$^3$ and used GP instead of PC in 0, 10, 20, 30, 40, and 50% in UHPC mixes (100–500 kg/m$^3$). The other groups were the same, but the second and third groups used 50% S + 50% G and 100% G, respectively. UHPC specimens were designed with a constant amount of SF, QP, steel fiber, SP, and a constant water/binder ratio, which were 200 kg/m$^3$, 150 kg/m$^3$, 117 kg/m$^3$, 26.4 kg/m$^3$, and 0.19, respectively. Three UHPC mixed series with two kinds of fine particles (S and G) and GP as pozzolanic materials were prepared based on previous studies [32,92]. The flow chart of the implemented experimental program is illustrated in Figure 4.

**Table 5.** Proportions of UHPC mixtures kg/m$^3$.

| Mixture ID | Portland Cement | Silica Fume | Quartz Powder | Glass Powder | Sand | Glass Particles | Steel Fiber | Superplasticizer | Water |
|---|---|---|---|---|---|---|---|---|---|
| PC100-S100 | 1000 | 200 | 150 | 0 | 701.5 | 0 | 117 | 26.4 | 228 |
| PC90 GP10-S100 | 900 | 200 | 150 | 100 | 683.6 | 0 | 117 | 26.4 | 228 |
| PC80 GP20-S100 | 800 | 200 | 150 | 200 | 665.8 | 0 | 117 | 26.4 | 228 |
| PC70 GP30-S100 | 700 | 200 | 150 | 300 | 647.9 | 0 | 117 | 26.4 | 228 |
| PC60 GP40-S100 | 600 | 200 | 150 | 400 | 630.0 | 0 | 117 | 26.4 | 228 |
| PC50 GP50-S100 | 500 | 200 | 150 | 500 | 612.2 | 0 | 117 | 26.4 | 228 |
| PC100-S50 G50 | 1000 | 200 | 150 | 0 | 347.9 | 347.9 | 117 | 26.4 | 228 |
| PC90 GP10-S50 G50 | 900 | 200 | 150 | 100 | 339.1 | 339.1 | 117 | 26.4 | 228 |
| PC80 GP20-S50 G50 | 800 | 200 | 150 | 200 | 330.2 | 330.2 | 117 | 26.4 | 228 |
| PC70 GP30-S50 G50 | 700 | 200 | 150 | 300 | 321.3 | 321.3 | 117 | 26.4 | 228 |
| PC60 GP40-S50 G50 | 600 | 200 | 150 | 400 | 312.5 | 312.5 | 117 | 26.4 | 228 |
| PC50 GP50-S50 G50 | 500 | 200 | 150 | 500 | 303.6 | 303.6 | 117 | 26.4 | 228 |
| PC100-G100 | 1000 | 200 | 150 | 0 | 0 | 690.9 | 117 | 26.4 | 228 |
| PC90 GP10-G100 | 900 | 200 | 150 | 100 | 0 | 673.3 | 117 | 26.4 | 228 |
| PC80 GP20-G100 | 800 | 200 | 150 | 200 | 0 | 655.7 | 117 | 26.4 | 228 |
| PC70 GP30-G100 | 700 | 200 | 150 | 300 | 0 | 638.2 | 117 | 26.4 | 228 |
| PC60 GP40-G100 | 600 | 200 | 150 | 400 | 0 | 620.6 | 117 | 26.4 | 228 |
| PC50 GP50-G100 | 500 | 200 | 150 | 500 | 0 | 602.9 | 117 | 26.4 | 228 |

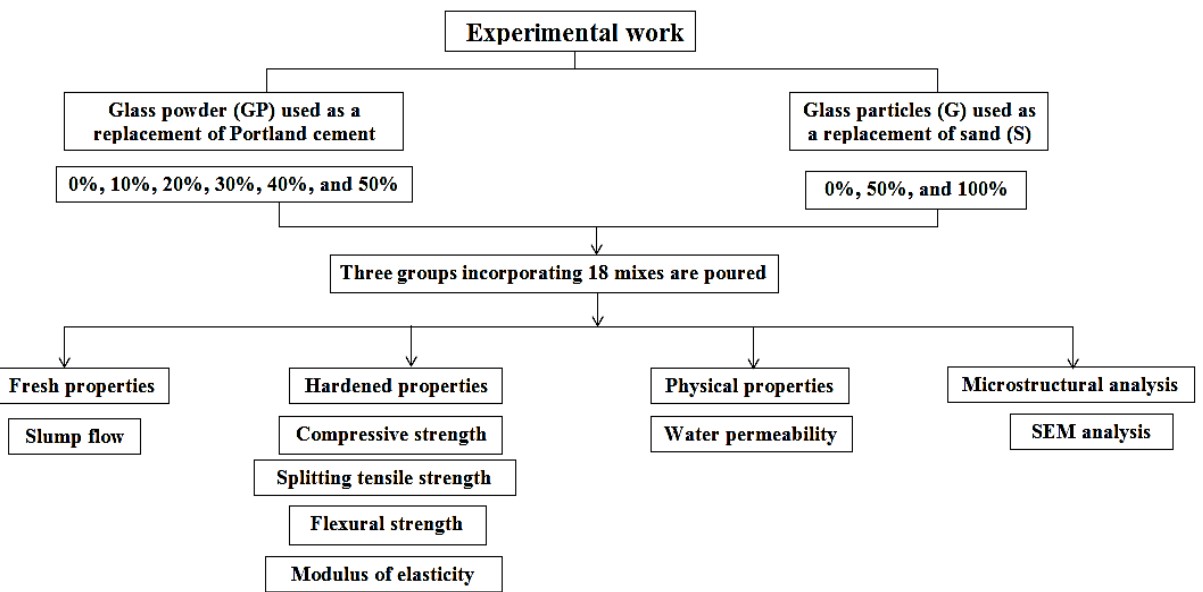

**Figure 4.** Flow chart of the implemented experimental program.

Fine aggregates and binder ingredients were added to a mixer and stirred for approximately three minutes or until the mixture was homogenous. During the mixing process, we progressively added ½ of the water and excess water together with the SP. Samples were cast and left covered in the casting hall for 24 h. After 24 h, the prepared samples were removed from the mold and kept in a curing chamber at 24 ± 2 °C as per ASTM C192 [93]. Then, these samples were water cured until the testing ages.

### 2.3. Test Procedure

Slump flow, compressive strength, splitting tensile strength, flexural strength, modulus of elasticity, and water permeability were the investigated characteristics of UHPC in this research, in addition to microstructural analysis. According to ASTM C, 143 [94], the slump flow test determines the workability of UHPC mixes (see Figure 5a). The compressive strengths at 1, 7, 28, and 91 days were determined on cubes of specimens (100 × 100 × 100 mm) (see Figure 5b), based on BS EN 12390-3 [95]. The recorded test results were the average of three measurements. The splitting tensile strength test at 28 days was determined based on ASTM C 496 [96]. Three cylindrical samples of dimensions 150 × 300 mm for each mix were utilized for this test at 28 days (see Figure 5c). The flexural strength of UHPC was evaluated based on the method described in ASTM C-78 [97]. Prism-shaped samples of 100 × 100 × 500 mm were used for the flexural strength tests (see Figure 5d). The modulus of elasticity at 28 days was determined using three cylinder-shaped (150 × 300 mm) UHPC samples (see Figure 5e) based on ASTM C-469 [98]. A water permeability test was conducted on cylindrical specimens of 15 × 15 cm at 28 days of age, according to BS EN 12390-8 [99]. Finally, an "SEM" examination was used to acquire the microstructure of the UHPC samples. Energy dispersive X-ray (EDX) analysis was performed. After 28 days of curing, the samples were sent to a lab at the Science Faculty at Alexandria University, Egypt, where they were analyzed for further findings.

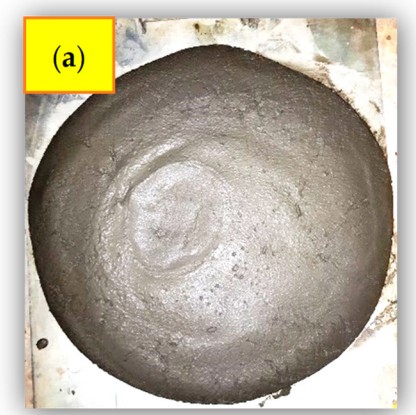
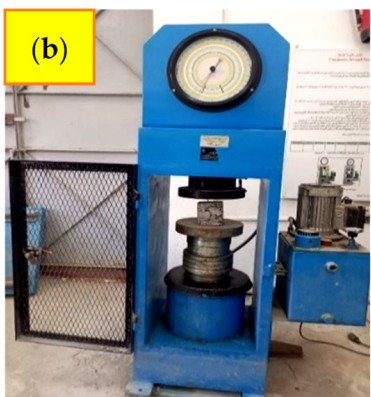
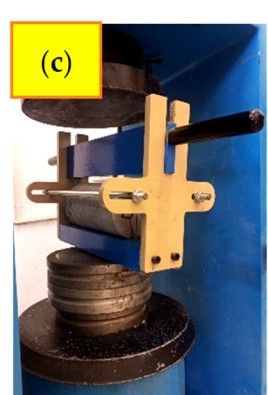
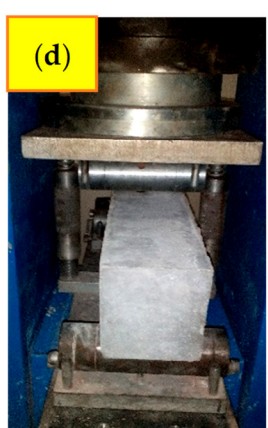
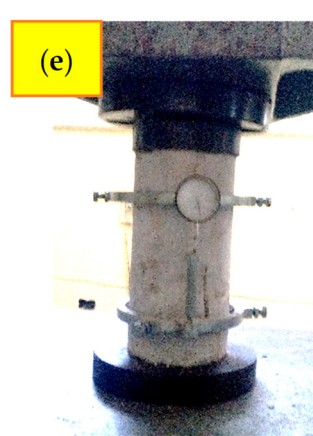

**Figure 5.** Testing procedures in this study: (**a**) slump flow, (**b**) compression test, (**c**) indirect splitting tensile test, (**d**) flexural test, and (**e**) modulus of elasticity test.

## 3. Results and Discussion

### 3.1. Slump Flow

All slump flow outcomes, represented in terms of flow diameter, for all UHPC mixes found in the three groups are shown in Figure 6. It was noted that the flow diameter rose as a result of raising the amount of G in the S:G ratio, implying enhanced workability. This result is in agreement with [66,68]. This might be because the glass absorbs less water than natural S, which increases the free water resulting in the mix. For clarification, the flow diameter values of mixtures PC100-S100, PC100-S50 G50, and PC100-G100 were 518 mm, 531 mm, and 540 mm, respectively. From a different viewpoint, it was discovered that when the replacement level of S with G is constant and the substitution rate of PC with GP is increased, the diameter of flow decreases, which implies reduced workability [100–102]. This may be due to the GP molecules being finer than PC molecules; the surface area of GP was 5630 cm$^2$/gm, while for PC it was 3495 cm$^2$/gm, see Table 1. Furthermore, because of the large surface area of GP, an extra amount of water was absorbed, and the free water was reduced [103]. For example, in the second group, the flow diameters of mixtures PC100-S50 G50, PC90 GP10-S50 G50, PC80 GP20-S50 G50, PC70 GP30-S50 G50, PC60 GP40-S50 G50, and PC50 GP50-S50 G50 were 531 mm, 522 mm, 513 mm, 505 mm, 496 mm, and 487 mm, respectively. Finally, the mixture PC100-G100 achieved the highest workability, recording a diameter flow of 540 mm, while the mixture PC50 GP50-S100 achieved the lowest workability, recording a diameter flow of 475 mm.

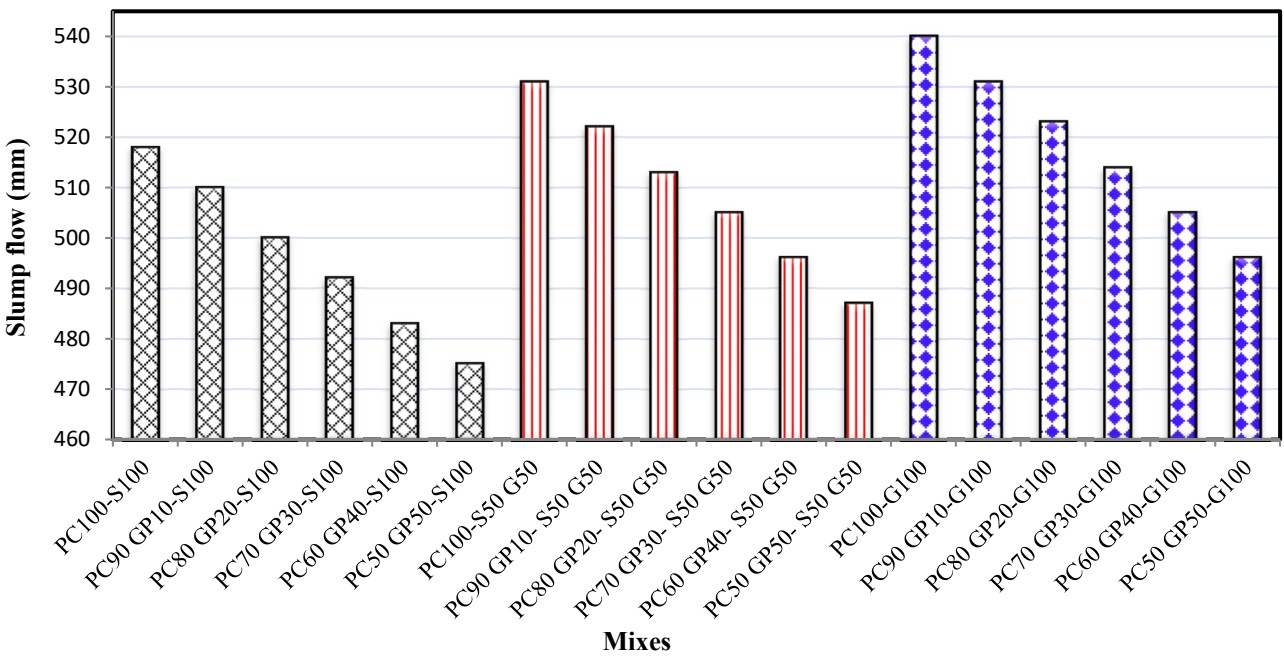

**Figure 6.** Slump flow of all UHPC mixtures.

*3.2. Mechanical Characteristics*

3.2.1. Compressive Strength

The compressive strength outcomes at several ages (1, 7, 28, and 91 days) of all the poured UHPC mixes are demonstrated in Figure 7. In general, the results illustrate that increasing the GP content as a partial replacement for PC from 0% up to 50%, while maintaining a constant ratio of S to G, has a negative impact on the UHPC compressive strength at a test age of 1 day implemented in this investigation. This finding is a familiar phenomenon for many supplementary cementitious materials during the replacement of large quantities of cement [51,104,105]. On the contrary, for the other tests, i.e., after 7, 28, and 91 days, it was noted that the compressive strength was enhanced up to a 20% replacement ratio, while after that, it reduced. The recorded compressive strengths after one day when utilizing GP as a partial substitution by 0%, 10%, 20%, 30%, 40%, and 50% of PC weight, in addition to 0% replacement of S with G (for clarification), were 70.4 MPa, 68.5 MPa, 66.8 MPa, 57.6 MPa, 50.3 MPa, and 44.6 MPa, respectively. While the recorded compressive strengths at 91 days in the same proportions were 188 MPa, 194.5 MPa, 201 MPa, 191.3 MPa, 185.2 MPa, and 181.4 MPa, respectively. This might be due to the lengthy curing time necessary to finish the calcium hydroxide reaction mechanism (CH). After 28 days, in the compressive strengths of the first group mixtures, it was noted that the mixtures PC100-S100, PC90 GP10-S100, PC80 GP20-S100, PC70 GP30-S100, PC60 GP40-S100, and PC50 GP50-S100 had compressive strengths of 167.8 MPa, 172 MPa, 176.3 MPa, 159.4 MPa, 151 MPa, and 141.8 MPa, respectively. However, the compressive strength values at the same test age for the second group mixtures (PC100-S50 G50, PC90 GP10-S50 G50, PC80 GP20-S50 G50, PC70 GP30-S50 G50, PC60 GP40-S50 G50, and PC50 GP50-S50 G50) were 163.2 MPa, 167 MPa, 171 MPa, 154.4 MPa, 145.9 MPa, and 137.7 MPa, respectively. Finally, the compressive strengths after 28 days for the third group, which included mixtures PC100-G100, PC90 GP10-G100, PC80 GP20-G100, PC70 GP30-G100, PC60 GP40-G100, and PC50 GP50-G100, were 155.6 MPa, 159.3 MPa, 163.1 MPa, 147.2 MPa, 139.1 MPa, and 131.3 MPa, respectively. From the above results presented for each group, it was noted that the compressive strength values increased up to a certain level, i.e., 20% replacement ratio of PC with GP, which is thus considered the optimum ratio to achieve the best compressive strength results. After this level it decreased, this might be due to several variables, including the fineness of GP, which aids in the enhanced activity of pozzolanic

cementitious materials (in agreement with [51–53]). Regarding replacement of S with G, adding more G to the mix in place of S is not advised. It was noted that boosting the substitution levels of S with G reduced the resulting compressive strength, which may indicate less attachment between the PC and glass than between natural S and the PC [104]. This might be because G absorbs less water than natural S, and because G is smooth, decreasing the adhesion between G and PC paste. This is in agreement with [64,66–71]. Finally, the other obtained compressive strength results at different ages (1, 7, and 91 days) are in agreement with the 28-day compressive strength findings.

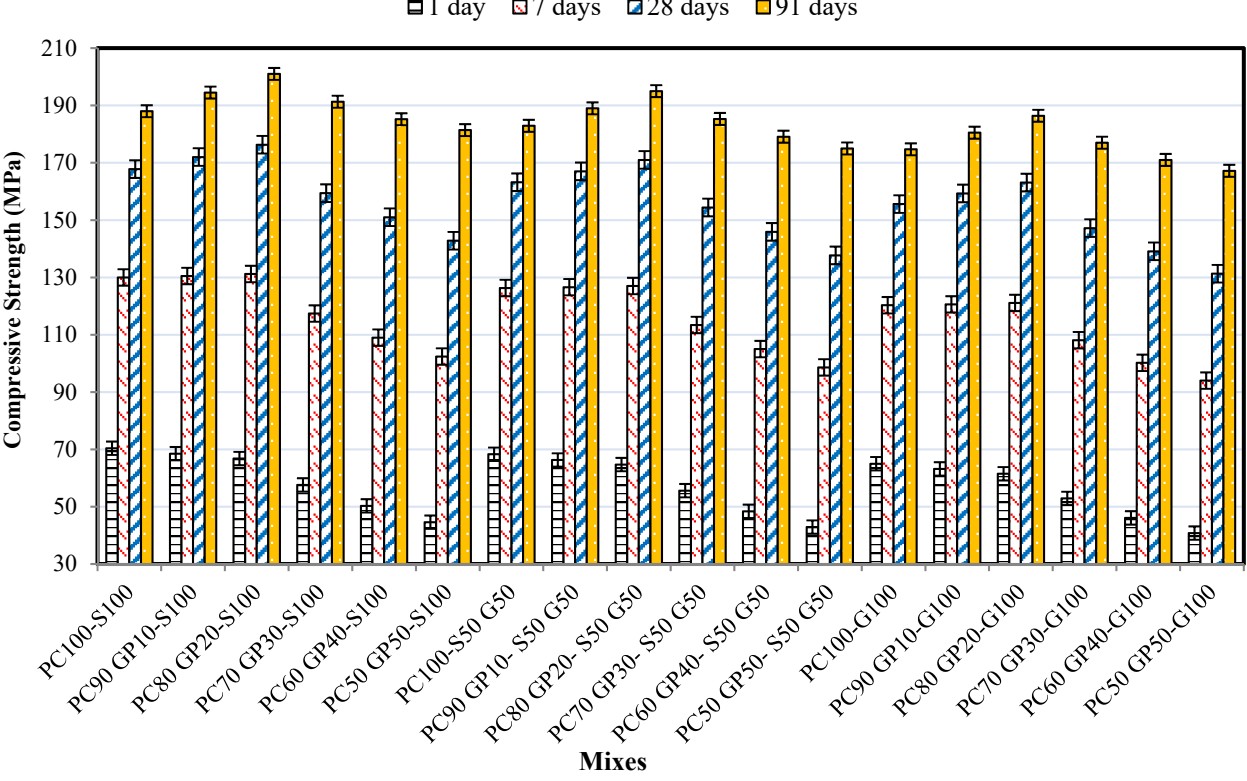

**Figure 7.** Compressive strength outcomes of all UHPC mix at various ages.

### 3.2.2. Splitting Tensile Strength

Figure 8 illustrates the splitting tensile strength outcomes of all UHPC mixes obtained at 28 days. It illustrates the relationship between the mixture name on the horizontal axis with the splitting tensile strength values in MPa on the vertical axis. For each group, it was noted that when the proportion of S was constant and the ratio of PC to GP was changed, the splitting tensile values increased up to 20% replacement, while after that, they decreased. These results are in agreement with [51,53]. For clarification, in the first group, with 0% replacement of G with S, in addition to replacing the PC content with ratios of GP of 10%, 20%, 30%, 40%, and 50%, the splitting tensile strengths were 16.9 MPa, 17.4 MPa, 18MPa, 16.4 MPa, 15.7 MPa, and 15 MPa, respectively. On the other hand, it was discovered that increasing the replacement ratios of S with G at a constant replacement ratio for PC/GP negatively affected splitting tensile values. This is in agreement with [64]. For example, the splitting tensile values for mixtures PC80 GP20-S100, PC80 GP20-S50 G50, and PC80 GP20-G100 were 18 MPa, 17.8 MPa, and 17.3 MPa, respectively. Finally, it was concluded that using 20% of GP as a replacement for PC with 0% replacement of S with G achieved the highest splitting tensile strength value [104]. Furthermore, all reasons which lead to these results are related to the same reasons mentioned in the results of the compressive strength.

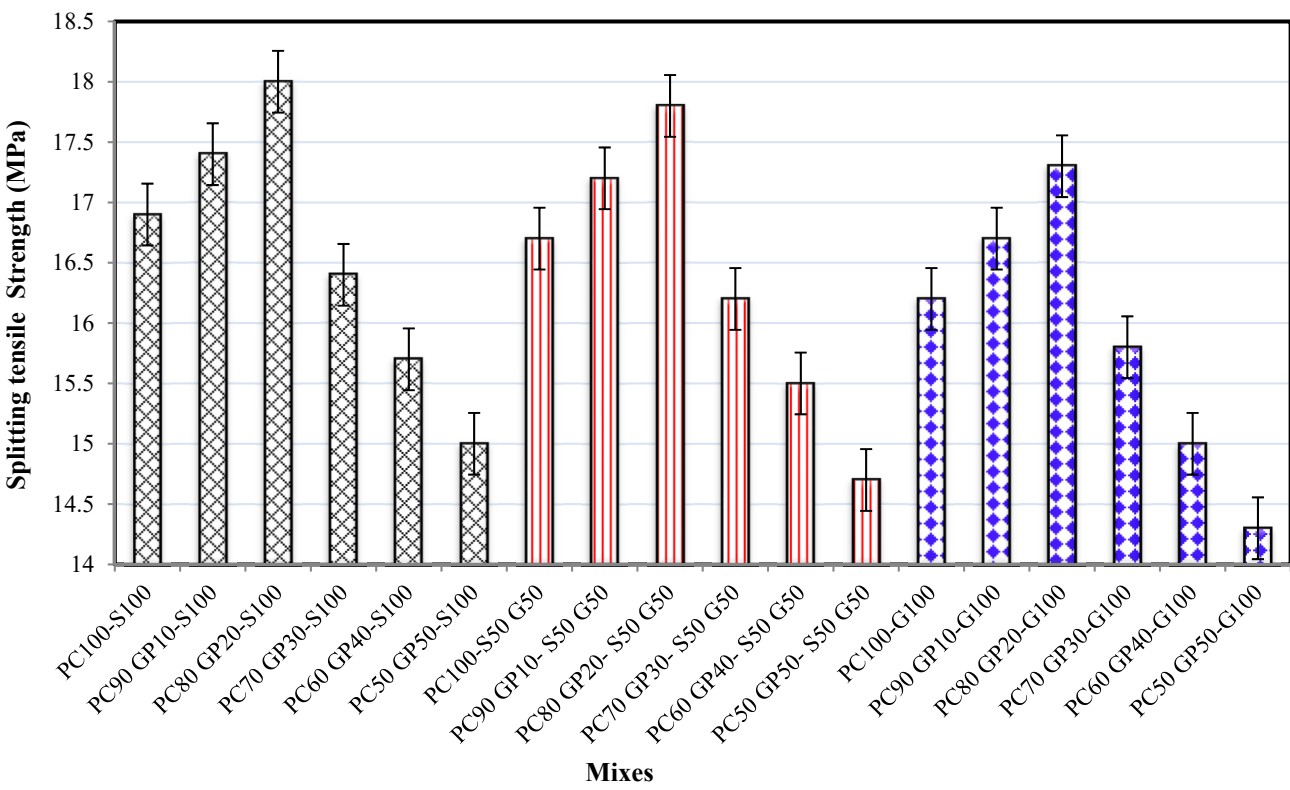

**Figure 8.** Splitting tensile strength results of whole UHPC mixtures at 28 days.

### 3.2.3. Flexural Strength

The obtained flexural strength results of all of the casted UHPC mixtures are shown in Figure 9, illustrating the impact of replacing PC with GP and S with G on the flexural strength determined at 28 days. In the first group, it was noted that the highest flexural strength value was achieved in the PC80 GP20-S100 mixture, at 25.7 MPa, which was an increase of about 6.19% when compared with the control mix in this group ("PC100-S100"). The lowest value was found in the "PC50 GP50-S100" mixture, at 21.3 MPa, which was a decrease of about 11.98% when compared with the same reference mix. From another point of view, it was discovered that the highest flexural strength value at the casted UHPC 18 mix occurred in the mixture "PC80 GP20-S100", at 25.7 MPa, while the lowest value occurred in the mixture PC50 GP50-G100, at 20.1 MPa. This means the optimum dosage to replace PC with GP and S with G is 20% and 0%, respectively [49]. This result agrees with various research [66,67]. The scientific explanation for this may be found in the compressive strength results section.

### 3.2.4. Modulus of Elasticity

In structural calculations, the elasticity modulus is classified as one of the essential parameters. Figure 10 shows the variety in the elasticity modulus of all the UHPC mixes in the three groups after 28 days. It was noted that there is a positive relationship between the elasticity modulus with the compressive strength. The outcomes revealed that the elasticity modulus values for the whole UHPC mix follow the same trends as the compressive strength values. The greatest modulus of elasticity is presented in the first group, at 57.82 GPa, which was accomplished at 20% substitution of PC with GP and 0% substitution of S with G. This value is followed by 57.03 GPa, which was achieved in the second group when using 20% substitution of PC with GP and 50% replacement of S with G; finally, the previous value is followed by 55.76 GPa in the third group, which was achieved when using 20% substitution of PC with GP and 100% replacement of S with G. That means the optimal value occurred at 20% substitution of PC with GP in addition to 0% replacement of

S with G. This result is in agreement with many research works [56,64]. Again, the main explanation for these results is found above in the compressive strength results.

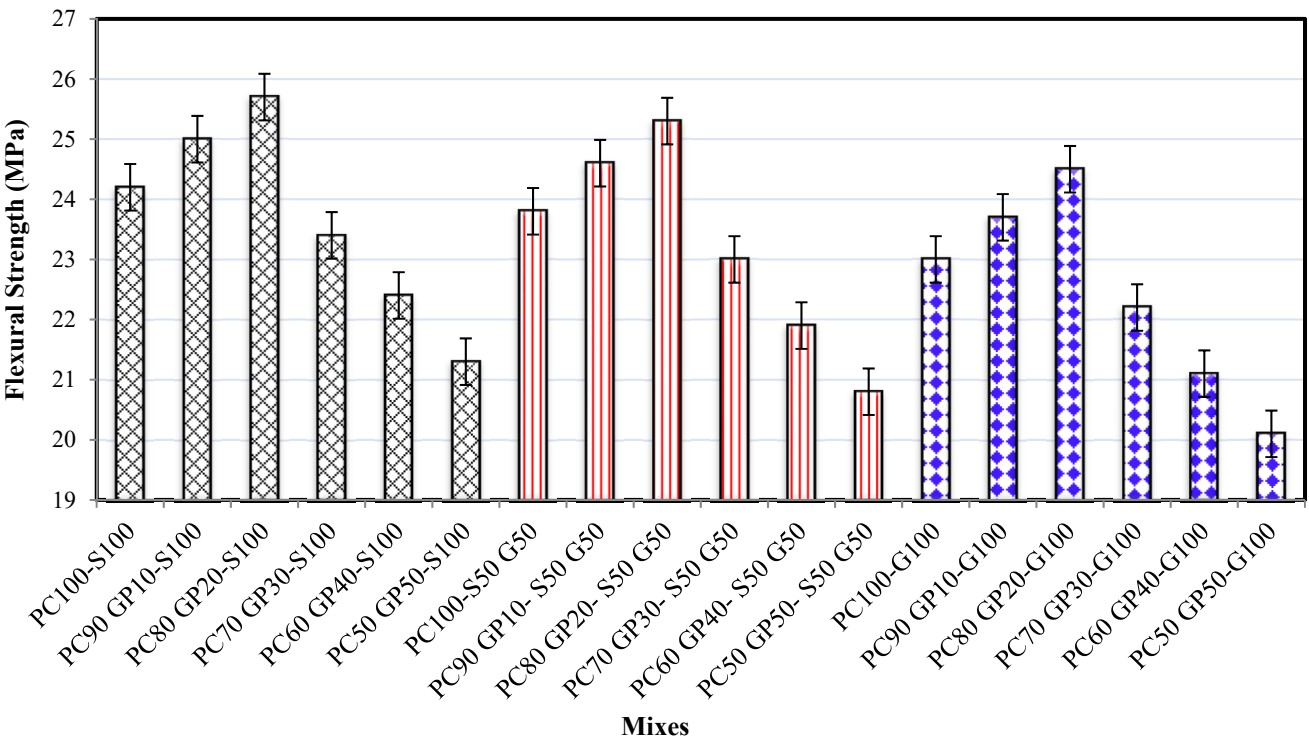

**Figure 9.** Flexural strength results of whole UHPC mixtures at 28 days.

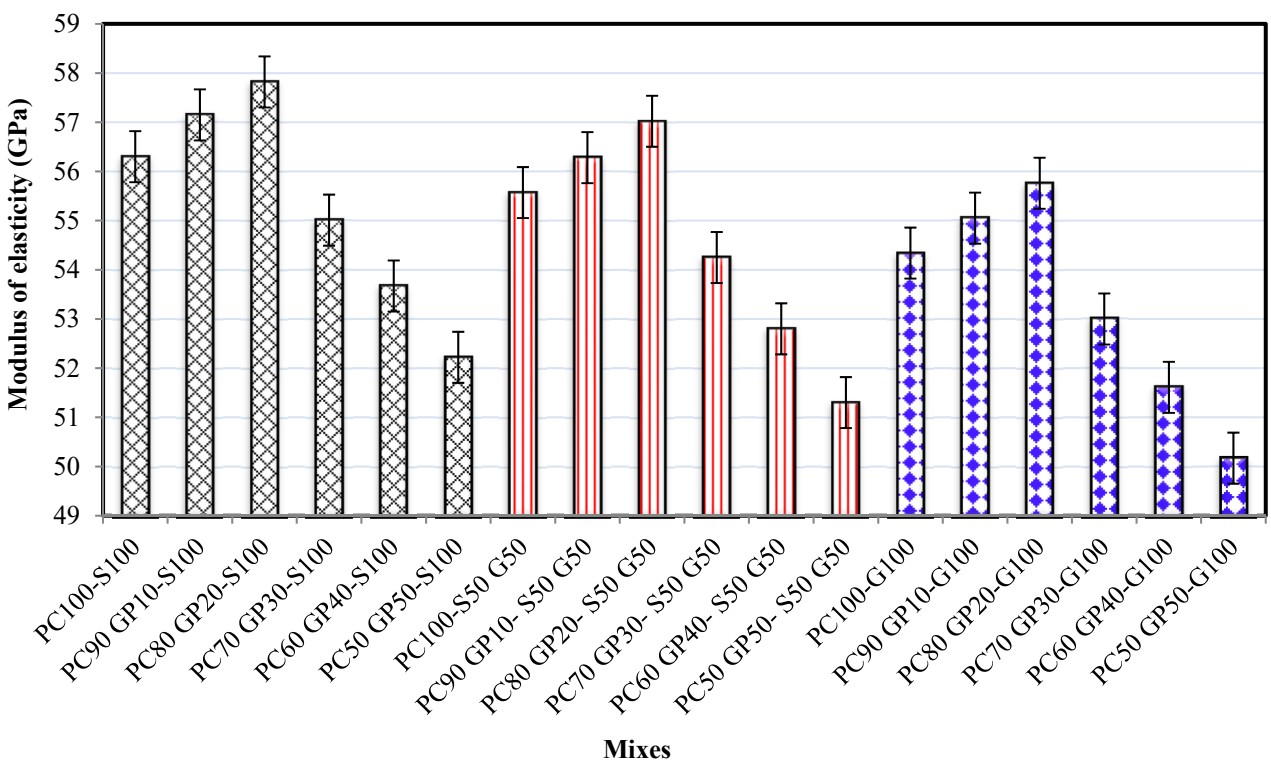

**Figure 10.** Modulus of elasticity outcomes of the whole UHPC mixes at 28 days.

### 3.3. Water Permeability

Figure 11 demonstrates the water permeability outcomes of UHPC mixtures after 28 days. Generally, it was noted that boosting the substitution ratio of GP to PC with a constant replacement ratio of G to S resulted in a remarkable reduction in water permeability. For clarification, in the first group, the water permeabilities of mixtures PC100-S100, PC90 GP10-S100, PC80 GP20-S100, PC70 GP30-S100, PC60 GP40-S100, and PC50 GP50-S100 were $1.44 \times 10^{-11}$ cm/s, $1.38 \times 10^{-11}$ cm/s, $1.28 \times 10^{-11}$ cm/s, $1.23 \times 10^{-11}$ cm/s, $1.17 \times 10^{-11}$ cm/s, and $1.12 \times 10^{-11}$ cm/s, respectively. These results may be related to the fineness of GP particles being greater than the fineness of PC particles, which helps in closing the pores and in forming a consistent density. The surface areas of GP and PC were 5630 cm$^2$/gm and 3495 cm$^2$/gm, respectively, see Table 1. These results are in agreement with [67]. From another point of view, it was discovered that boosting the substitution rate of G with S with a constant substitution ratio of GP to PC increased the water permeability values. For example, the water permeability values of mixtures PC100-S100, PC100-S50G50, and PC100-G100 were $1.44 \times 10^{-11}$ cm/s, $1.48 \times 10^{-11}$ cm/s, and $1.53 \times 10^{-11}$ cm/s, respectively. This may reflect that the water absorption (%) value of S is higher than the water absorption (%) of G; the water absorption values of S and G are 0.85% and 0.65%. Additionally, S has a higher content of clay and fine materials than G, i.e., the clay and fine materials (%) of S is 0.47% while for G it is 0.30%. These materials have a great ability to absorb water, see Table 3. These outcomes agree with [82].

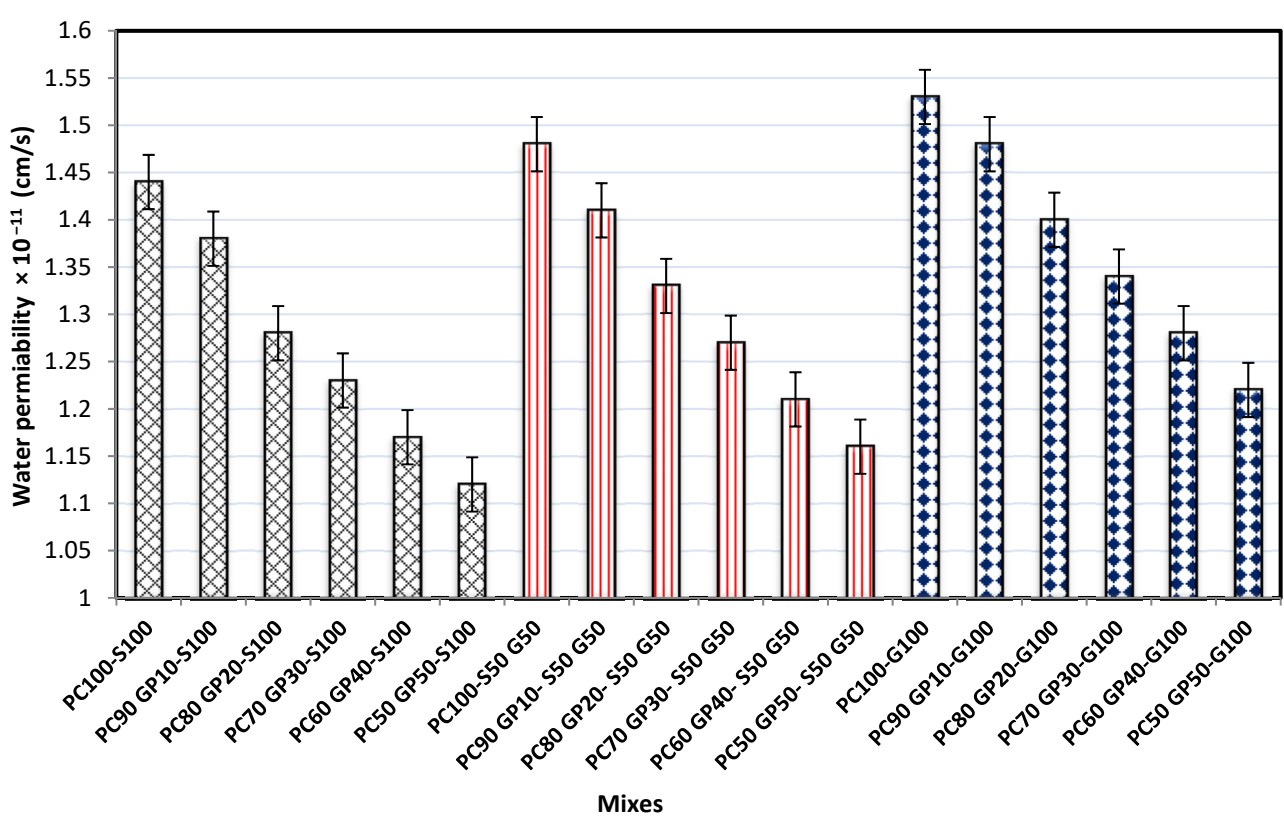

**Figure 11.** Water permeability outcomes of UHPC mixtures after 28 days.

### 3.4. Microstructure

The morphologies from the SEM images of the UHPC samples PC100-S100, PC90 GP10-S100, PC80 GP20-S100, PC70 GP30-S100, PC60 GP40-S100, and PC50 GP50-S100 are shown in Figure 12a–f. The micrographs of PC90 GP10-S100 and PC80 GP20-S100 (see Figure 12b,c) seem to be dense and more compact compared to the erratic and wavy micrographs of PC100-S100, PC70 GP30-S100, PC60 GP40-S100, and PC50 GP50-S100 (see Figure 12a,d–f),

which are characterized by their lack of response specimen and their interfacial fractures. These imperfections point to insufficient hydration and the microstructure's flakiness. As a result, the resulting mixture has a hydraulic ability which is capable of interacting with Ca(OH)$_2$ to form more C-S-H gel with 10% and 20% replacement of PC with GP, resulting in the best mixes with the optimal microstructure (thick with no gaps or fractures) and the best mechanical properties. This finding is in line with other research that has shown that concrete's engineering qualities are enhanced due to the material's high fineness and balanced chemical composition [105].

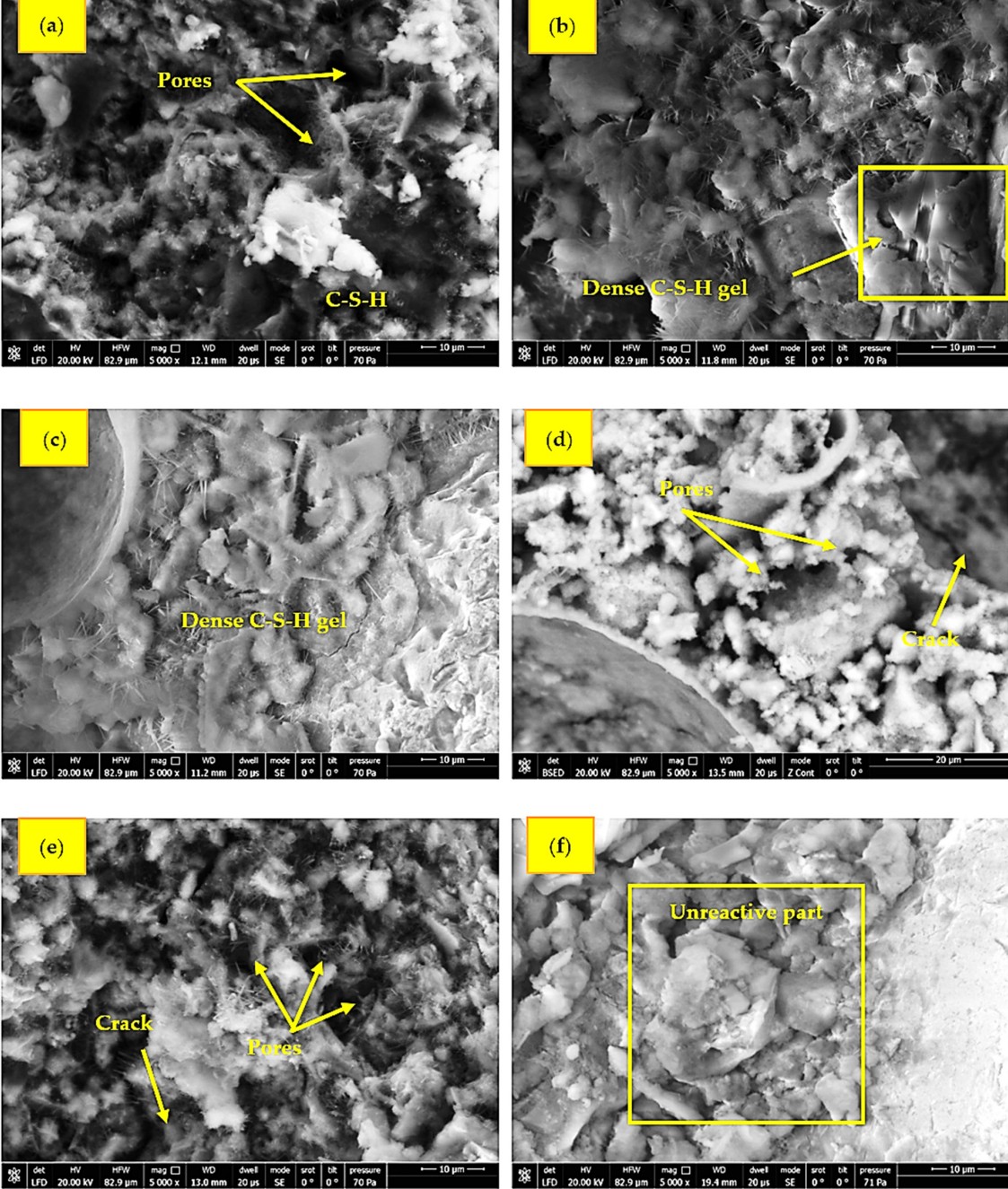

**Figure 12.** Micrographs of SEM of mixes (**a**) PC100-S100, (**b**) PC90 GP10-S100, (**c**) PC80 GP20-S100, (**d**) PC70 GP30-S100, (**e**) PC60 GP40-S100, and (**f**) PC50 GP50-S100.

The SEM images of the interfacial transition zone (ITZ) between the glass particles and cement paste at replacement levels of 0%, 50%, and 100% glass are shown in Figure 13a–c.

These levels correspond to PC100-S100, PC100-S50 G50, and PC100-G100 mixtures, respectively. Compared to mixtures PC100-S100 and PC100-S50 G50 (see Figure 13a,b), the poor ITZ between glass particles and the cement paste in the PC100-G100 mix is shown in Figure 13c. This may be because the adhesion is particularly poor to glass particles because of their smooth surface [13].

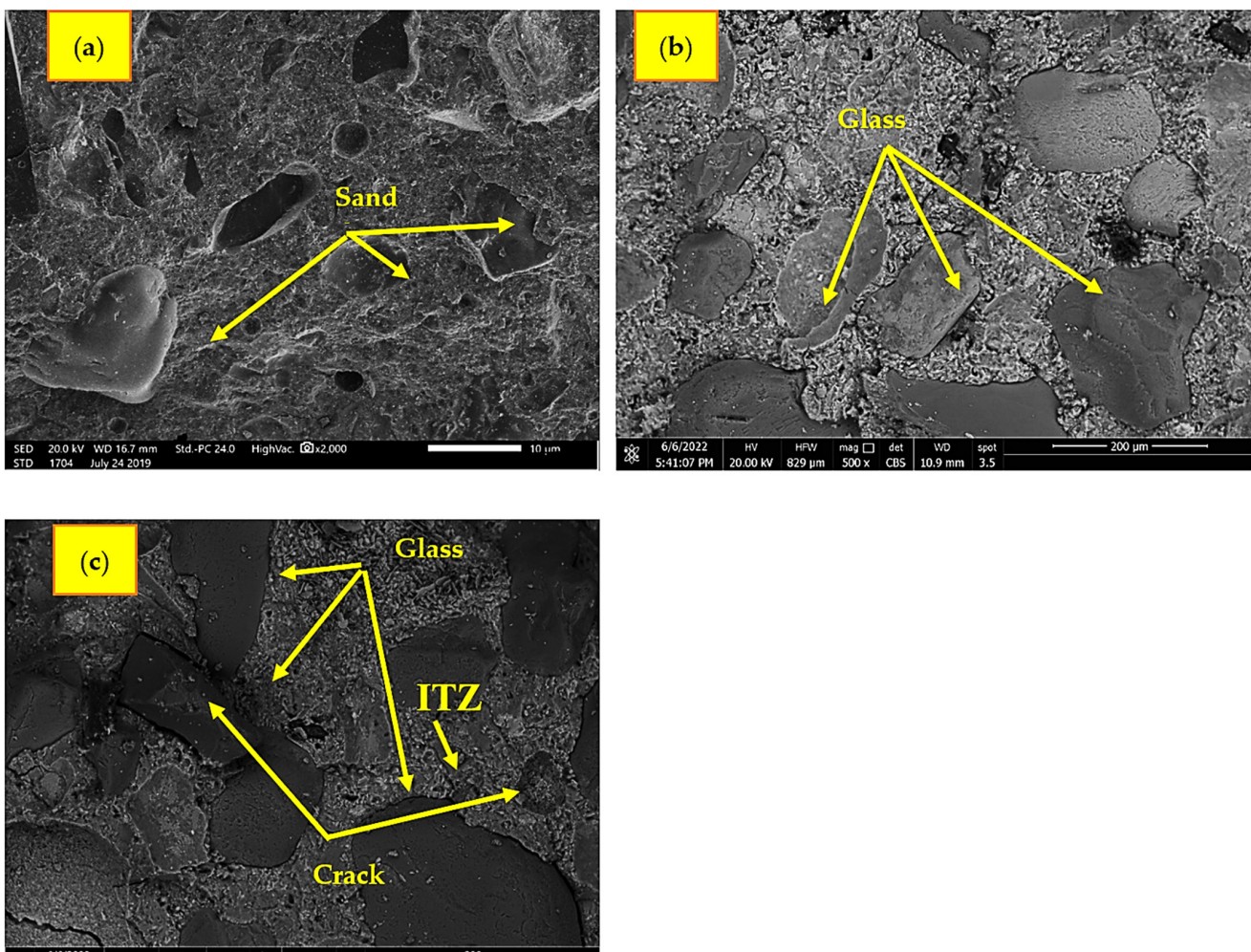

**Figure 13.** Micrographs of SEM of mixes (**a**) PC100-S100, (**b**) PC100-S50 G50, and (**c**) PC100-G100.

### 4. Conclusions

This research aims to produce an environmentally friendly and economical UHPC by utilizing GW with different sizes as a substitution for PC. The substitution levels investigated were 0%, 10%, 20%, 30%, 40%, and 50% using GP and for fine aggregates, the substitutions ratios were 0%, 50%, and 100% using G. To achieve this goal, 18 mixtures are made and tested experimentally. Fresh and hardened water permeability and microstructural characteristics are investigated. The following outcomes are concluded:

1. Due to raising the substitution level of S with G, the workability of UHPC is enhanced. On the other hand, raising the replacement ratio of PC with GP reduces the workability of UHPC when the replacement ratio of S with G is constant;

2. The optimum replacement ratios for PC with GP and S with G to manufacture an environmentally friendly and economical UHPC with high mechanical characteristics are 20% and 0%, respectively;

3. Increasing the GP content in UHPC as a partial replacement for PC from 0% up to 50%, with a constant replacement ratio of S with G, has a negative impact on the

compressive strength of UHPC after 1 day. On the contrary, after 7, 28, and 91 days it is noted that the compressive strength increases up to 20% replacement ratio while after that, it decreases;

4.  For all investigated mixtures of UHPC, the maximum obtained values for compressive strength, splitting tensile strength, flexural strength, and modulus of elasticity at 28 days were 176.3 MPa, 18 MPa, 25.7 MPa, and 57.82 GPa, respectively, when 20% replacement of PC with GP in addition to 0% replacement of S with G is implemented;

5.  Raising the replacement ratio of GP with PC, while keeping the substitution level of G with S constant, results in a remarkable reduction in water permeability. However, boosting the substitution level of S with G, with a constant GP to PC ratio, increases the water permeability values and leads to a notable reduction in drying shrinkage values.

## 5. Recommendations for Future Studies

For future studies, it is recommended to study the effect of elevated temperature on UHPC containing GP and G. In addition, investigations into the structural behavior of this type of material under static load, impact load, and cyclic load should be undertaken. Finally, a numerical analysis should be undertaken to investigate the structural behavior of UHPC containing GP and G.

**Author Contributions:** Conceptualization, M.H.A.-E., I.S.A., M.A., S.M. and N.M.; methodology, M.H.A.-E., I.S.A., M.A., S.M. and N.M.; software, M.H.A.-E., I.S.A., M.A., S.M. and N.M.; validation, M.H.A.-E., I.S.A., M.A., S.M. and N.M.; formal analysis, M.H.A.-E., I.S.A., M.A., S.M. and N.M.; investigation, M.H.A.-E., I.S.A., M.A., S.M. and N.M.; resources, M.H.A.-E., I.S.A., M.A., S.M. and N.M.; data curation, M.H.A.-E., I.S.A., M.A., S.M. and N.M.; writing—original draft preparation, M.H.A.-E., I.S.A., M.A., S.M. and N.M.; writing—review and editing, M.H.A.-E., I.S.A., M.A., S.M. and N.M.; visualization, M.H.A.-E., I.S.A., M.A., S.M. and N.M.; supervision, M.H.A.-E., I.S.A., M.A., S.M. and N.M.; project administration, S.M.; funding acquisition, S.M. All authors have read and agreed to the published version of the manuscript.

**Funding:** This research received no external funding.

**Institutional Review Board Statement:** Not applicable.

**Informed Consent Statement:** Not applicable.

**Data Availability Statement:** Not applicable.

**Conflicts of Interest:** The authors declare no conflict of interest.

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
