# Peer review of "Investigation of the Physical Mechanical Properties and Durability of Sustainable Ultra-High Performance Concrete with Recycled Waste Glass"

_sustainability, doi:10.3390/su15043085_

Round 1

Reviewer 1 Report

Title:

"Investigation of the physical-mechanical properties and durability of sustainable ultra-high-performance concrete with recycled waste glass"

The authors did a great job of researching various properties for sustainable ultra-high performance concrete containing wasted glass material. This is a novel topic in the field. The reviewer recommends the paper for publication in the "Sustainability" journal after making the following major modifications to bring it to the desired stage:

1.      In abstract, it is preferable to add the most important obtained outcomes

2.      The introduction is a bit weak, add more details about recycled waste glass.

3.      It is recommended to add subtitle 1.1 Research significance

4.      What does your study add to the subject area compared with other published material? Answer this question more clearly in the “Research significance” section.

5.      state the mix design procedure in the design of the poured mixes in this research

6.      What specific improvements should that future researchers consider regarding the methodology? What further controls should be considered?

7.      It is preferred  in figure "4" the vertical axis started from 460 mm instead of 450 mm

8.      Add a discussion section on comparison with previous studies.

9.      The manuscript needs minor language revision.

10.  Revise your conclusions in a way that is consistent with the evidence and arguments presented and does it address the main question at hand?

11.  It is preferable to add a section "Recommendations" to indicate the authors' recommendations for future researchers.

12.  Add references from recent years.

Author Response

Dear Reviewer,

Thanks for all the time and effort you put into reviewing our paper. This is a great contribution to the scientific community. It's much appreciated.

Regarding the comments, you will find the authors' responses in the attachments. Also highlighted in the revised manuscript.

Kind Regards,

Reviewer 2 Report

In this manuscript, the authors aim to produce a type of UHPC by using GW of different sizes as a substitute for PC at six proportions and G as a substitute for natural fine aggregate S at substitution rates of 0%, 50%, and 100%. The authors designed and tested 18 mixtures. The mechanical properties, water permeability, and microstructure of fresh and hardened concrete were investigated. The subject of the experiment may be of interest to the community. However, there are basic problems such as unclear expressions, contradictions between the diagram and text, and too shallow analysis of macroscope experimental conclusions. The internal mechanisms responsible have not been well explored and analyzed in conjunction with microscopic experiments. Therefore, this manuscript cannot be recommended for publication.

Specific comments:

1. Page 1, line 33: The abstract is incomplete. It should include content related to microstructures.

2. Page 2, lines 72-74: S appears for the first time in the Introduction section; please clearly express its meaning.

3. Page 3, lines 109-145: There are too many introductions to GW replacing natural aggregates, and the logic is unclear.

4. Page 4, lines 169-177: The specific process flow of using glass waste (GW) to produce glass particles (G) and glass waste powder (GP) with different particle sizes should be introduced in more detail, i.e., the specific instrument used, the mesh of the sieving hole, and the specific time of grinding.

5. Page 6, line 195: Fig. 2. In the Introduction section, it has been introduced that in most experiments, the GP size is less than 40 μm to obtain some activity. The GP activity of 75 μm particle size is not obvious. The basis for using 75 μm particle size in this manuscript should be provided.

6. Page 8, line 236: What does SLA represent in PC90 SLA10-S100 in Table 5?

7. Page 8, line 236: Why does the amount of natural fine aggregate (S) decrease when the proportion of GP replacing cement increases in the mix design? In this case, how to compare the mix proportions?

8. Page 10, lines 296-298: The 1d compressive strength of the three groups of mixtures decreases with the increase in GP substitution, and the 7d compressive strength basically remains unchanged when the GP substitution ratio is less than 20%, and then gradually decreases, which is inconsistent with the trend for the 28d compressive strength. The relevant discussion and analysis should be provided.

9. Page 11, lines 312-317: It is found that under the same replacement rate of PC, increasing the replacement rate of G from S has a negative effect on the splitting tensile value. The splitting tensile values of mixtures PC80 GP20-S100, PC80 GP20-S50 G50 and PC80 GP20-G100 are 18 MPa, 17.8 MPa and 17.3 MPa, respectively. However, the loss in splitting tensile strength caused by replacing 100% of S by G is less than 5%. Is this an indication that replacing S with G has a particularly low effect on mechanical strength loss?

10. Page 14, lines 357-360: Though Table 1 shows that the specific area of PC is lower than that of GP, it is known that the particle size of cement is generally less than 40 μm. In this manuscript, GP is ground at a scale of less than 75 μm, but it is concluded that the fineness of GP particles is greater than that of PC particles, which helps to seal pores and form a dense consistency. It seems contradictory.

11. Page 15, lines 372-375: The text here is inconsistent with the title of Fig. 10.

12. Page 15, lines 371-380: Why not analyze the microstructure of the mixture of GP replacing PC? The micro test results should be compared with the previous macro test results for analysis and discussion, such as slump and mechanical properties.

Author Response

(The authors gave the same response as above.)

Reviewer 3 Report

This paper shows an investigation on the properties of UHPC blended with recycled glass powder. Overall, this paper is well organized and can be considered after the following comments being addressed.

(1) The effects of recycled glass powder on the properties of blended UHPC have been investigated by many previous studies. The author should clear the shortage of previous investigations and the innovation of this study.

(2) The cited references are outdated, and the literatures on sustainable and recycled concrete in recent 3 years should be cited. The following one reference may be helpful. (a) Characterization of sustainable mortar containing high-quality recycled manufactured sand crushed from recycled coarse aggregate (https://doi.org/10.1016/j.cemconcomp.2022.104629).

(3) Line 51, “3” should be placed at superscript.

(4) Line 77, Ca(OH) or Ca(OH)2?

(5) If possible, the particle size and XRD pattern of glass powder as cement placement should be given.

(6) The author should give a comparison on the findings in this work and related literatures.

Author Response

(The authors gave the same response as above.)

Round 2

Reviewer 1 Report

The authors addressed comments as requested. Therefore, it can be accepted for publication.

Author Response

Thank you very much for all the feedback provided by reviewer #1. Consideration of the comments has improved the quality of our manuscript.

Reviewer 2 Report

This manuscript has been revised to a certain extent and may be published in the journal Sustainability.

Author Response

Thank you very much for the observation made by the reviewer #2. Consideration of the comments would improve the quality of our manuscript. The paper was modified based on the reviewer's comments, as it was grammatically revised and the overall structure of the manuscript was reviewed.